# The Effect of Nitridation on Sputtering AlN on Composited Patterned Sapphire Substrate

**DOI:** 10.3390/ma16031104

**Published:** 2023-01-27

**Authors:** Yi Zhang, Guangmin Zhu, Jiangbo Wang, Zichun Le

**Affiliations:** 1Institute of Optical Engineering, College of Science, Zhejiang University of Technology, No. 288, Liuhe Road, Hangzhou 310023, China; 2Key Laboratory of Wide Bandgap Semiconductor Materials and Devices, HC Semitek Corporation, No. 233, Suxi Road, Yiwu 322009, China

**Keywords:** composited patterned substrate, GaN epitaxy, sputtering AlN, light-emitting diode

## Abstract

Here, we report on the epitaxial growth of GaN on patterned SiO_2_-covered cone-shaped patterned sapphire surfaces (PSS). Physical vapor deposition (PVD) AlN films were used as buffers deposited on the SiO_2_-PSS substrates. The gallium nitride (GaN) growth on these substrates at different alternating radio frequency (RF) power and nitridation times was monitored with sequences of scanning electron microscopy (SEM) and atomic force microscopy (AFM) imaging results. The SEM and AFM show the detail of the crystalline process from different angles. Our findings show that the growth mode varies according to the deposition condition of the AlN films. We demonstrate a particular case where a low critical alternating current (AC) power is just able to break SiO_2_ covalent bonds, enabling the growth of GaN on the sides of the patterns. Furthermore, we show that by using the appropriate nitridation condition, the photoluminescence (PL) integral and peak intensities of the blue light epi-layer were enhanced by more than 5% and 15%, respectively. It means the external quantum efficiency (EQE) of epitaxial structures is promoted. The screw dislocation density was reduced by 65% according to the X-ray diffraction (XRD) spectra.

## 1. Introduction

GaN light-emitting diode (LED) technology has witnessed significant developments in the past three decades. The external quantum efficiency (EQE) has been greatly improved to above 80% (blue) and 60% (green) by applying newly developed epitaxial fabrication technologies to industrial products. Meanwhile, the substrates are the foundations of the GaN films, greatly affecting their crystalline qualities and optical paths. Owing to their advantageous properties, including light transmittance, low cost, and strong hardness, c-plane sapphire substrates have been widely used in the mass production of GaN-based devices [1]. However, dislocations induced by strain relaxation and caused by the mismatch of lattice constants between sapphire and GaN, are unavoidable [2,3,4,5]. Considerable efforts have been made to reduce the negative effects of lattice mismatches on the growth of GaN films. A recent approach based on the use of the epitaxial lateral overgrowth (ELOG) technique in conjunction with a new substrate design, known as the patterned sapphire substrate (PSS), has been investigated. The new PSS design can enhance the ELOG process, eliminating the dislocations by changing the trends of dislocation lines [6,7,8,9,10,11,12,13]. The epitaxial lateral overgrowth process is very necessary to the elimination of screw dislocations which start from the c-plane of sapphire. When there is no pattern on the sapphire surface, the screw dislocations extend vertically through the GaN layer, leading to electron leakage passages and nonradiative recombination centers. The incline surfaces of PSS hardly produce screw dislocations and force the dislocation lines to turn to tilt directions, to confluence, and finally to vanish.

Furthermore, the differences in the refractive indices of GaN (*n* = 2.45), sapphire (*n* = 1.76), and air (*n* = 1) cause a total internal reflection effect, limiting the extraction of light from InGaN/GaN quantum wells (QWs). Longer light paths lead to more absorption by QWs and shallow energy levels induced by defects and impurities. Therefore, engineering shape designs on the substrate can contribute to higher light extraction efficiencies (LEE) by changing the emission patterns and shortening light paths [13,14]. The cone-shaped PSS is widely used for the elaboration of GaN-based blue and green LED applications, while ultra-violet LED applications mostly use pit-shaped PSS [15,16]. By replacing the material in the cones with silicon oxide (*n* = 1.45), the LEE can be further enhanced as it becomes harder for the light to penetrate through the substrate [17]. The LEE promotion is achieved with the total reflection enhancement. The n value of silicon oxide is smaller than that of sapphire, leading to more total reflections at the interface between SiO_2_ and sapphire. It is a very significant effect for the tilt light to be reflected to the PGaN on top of the epitaxial layers instead of penetrating through the sapphire substrate. Generally speaking, it leads to more light being vertically reflected on the GaN surface and being extracted directly into the air instead of being absorbed by the surface levels on the sides.

The magnetron sputtering physical vapor deposition (PVD) process is a mature technology that has long been applied to the LED epitaxial procedure [18,19,20]. Better EQE and reduced costs were attained using sputtered AlN films deposited by PVD as buffers instead of GaN films deposited by metal/organic vapor phase epitaxy (MOVPE). Despite its small thickness, the PVD-deposited buffer AlN layer can significantly affect the quality of GaN layers [21,22,23,24]. The versatility of the sputtering process enables the nitridation of substrates using radio frequency (RF) alternating electric fields [25]. Their power and duration are key factors influencing the nitridation process. In this study, our attention is turned to the crystal lattice mismatch between AlN and c-plane sapphire, which is larger than that between AlN and SiO_2_. Our interest is focused on the mechanism by which the AlN buffer cooperates with the SiO_2_-patterned PSS, leading to a higher quality GaN. The mature LED epitaxial structure design is guided by the theoretical work to a great extent as well [26,27,28,29].

To investigate this mechanism, we prepared sets of GaN samples on AlN films with different deposition conditions to study the epitaxial process.

## 2. Experiments

The SiO_2_-patterned sapphire substrates were fabricated using inductively coupled plasma (ICP) from flat substrates with SiO_2_ films deposited by plasma-enhanced chemical vapor epitaxy (PECVD). The hexagonal periodic arrangement patterns were changed into cone shapes of the same size: approximately 2.8 μm wide and 1.7 μm high. The periodic length was 3.1 μm. The sapphire parts under the SiO_2_ films were etched to a 300 nm depth for the further nucleation of GaN in the next steps.

We prepared 32 pieces of 4-inch substrate divided into 8 groups. For each group, AlN films were deposited on the substrates in the same chamber under the same conditions (see Table 1). An additional flat c-plane sapphire substrate was incorporated into the chamber for each group deposition. All the sputtering processes were performed at 400 °C with a 1:1 N_2_ to Ar ratio. Subsequently, a new sample arrangement was made by collecting sets of 8 substrate templates with totally different AlN films into separate groups, ready for the epitaxial steps in the MOVPE chamber. Two steps remained to complete the growth of intrinsic GaN as the bottom layer, similar to the usual methods applied to flat substrates. First, a 600 nm layer was deposited with 500 torrs at 1000 °C for the nucleation, followed by a 1000 nm layer with 100 torrs at 1150 °C for the ELOG. Four nodes were recorded during the bottom layer growth (i.e., 300 nm, 600 nm, 1000 nm, and 1600 nm) using atomic force microscopy (AFM) and scanning electron microscopy (SEM) imaging.

Furthermore, the epitaxial LED structures were completed on another set of 8 templates in a single run. The structure’s architecture comprises the intrinsic GaN layer mentioned above, a 1 μm heavy doped N-type GaN layer at 1100 °C, 300 nm superlattice layer for the strain relaxation at 900 °C, 150 nm multi-quantum wells with 12 pairs superlattice (barrier at 850 °C/well at 780 °C), and a 300 nm heavy doped P-type GaN layer at 900 °C. An additional sample was grown on a normal sapphire substrate with the same size of patterns. It is worth noting that the epitaxial processes were conducted in different runs to obtain the same wavelength as the thermal expansion coefficients are very different between SiO_2_ (0.5 × 10^−6^/K) and Al_2_O_3_ (7.5 × 10^−6^/K) [30,31]. Finally, HD-XRD was used to characterize the structural properties of the obtained samples, and their PL spectra were acquired at room temperature to characterize the optical response characteristics. 

## 3. Results and Analysis

Figure 1 shows SEM images of the bottom layer’s growth processes using the conditions listed in Table 1. The nitridation duration for samples 1–4 was fixed at 60 s and the AC powers were varied between 50 and 120 W. For samples 5–7, the AC power was fixed at 50 W while varying the nitridation duration between 90 and 150 s. No nitridation process was applied to sample 8 for contrast reference. 

The images recorded for the four thickness values illustrate the four representative moments during the crystal growth. They offer a view of the growth mechanism at different stages as follows. For sample 1, the images at the 300 nm thickness show an already completed nucleation, where the growth is observed to start along the edges, a little over the SiO_2_ part of the patterns. At this stage, the patterns should remain sharply circular from the top view. The 600 nm images show the emergence of the GaN hexagonal structure, where the cone-shaped patterns were still partially uncovered. This indicated that the ELOG process forced by the patterns was about to complete. The 1000 nm images show the moment when the patterns had been completely covered, while the large screw dislocations originating from the top of the patterns remained. Each dark dot on the images corresponds to one piece of the pattern. The 1600 nm images show the moment when those dislocations had been eliminated or transformed into small screws resulting in smooth surfaces (i.e., *R_a_* < 0.2 nm).

The growth mechanisms before the thickness reached 300 nm were changing with the AC power varying from 70 W (sample 2) to 90 W (sample 3). The surrounding GaN crystal adopts remarkably irregular structures compared to those observed in sample 1. Specifically, samples 3 and 4 turn to flower-shaped surfaces instead of the expected hexagonal structures. The analysis of the AFM cross-section profiles shown in Figure 2 provides additional insight into the growth mechanisms. Usually, the nucleation process began on the flat c-plane of sapphire, AlN, or GaN. In this circumstance, the only places which could be chosen to grow are those bottom areas between the patterns. The crystal “islands” continue expanding into surrounding spaces and the surface remains irregular for some time, matching the regular bulges between the patterns, similar to the mechanism observed in sample 1. However, those bulges seem to be some kind of one-sided covers in samples 3 and 4. The growth of wurtzite GaN on the c-plane sapphire was disturbed and turned to grow on the sides of the patterns. Considering that the same variation of the AC power did not result in the behavior on the normal PSS, this phenomenon could then be attributed to the original properties of SiO_2_. The fact that the bond energy of Si-O is lower than that of Al-O and Al-N suggests that the SiO_2_ structure is more prone to break under the effect of the charged plasma particles compared to the Al_2_O_3_ structure. Noticeable trigonal crystal clusters (marked in red in Figure 2), extending in a petal-like fashion from the patterns, exhibit a strong resemblance to the SiO_2_ diamond-style crystal structure. It reveals that the GaN cluster grew on both the c-plane sapphire at the bottom and the SiO_2_ on the sides. It can be concluded that, although there is a big mismatch of approximately 8% between the SiO_2_ and GaN lattices, the latter can still grow on some of the planes of SiO_2_. The plasma created by the critical power can etch it gently and forces the appropriate planes to expose it. There are many advantages to using SiO_2_ as a substitute for Al_2_O_3_ for the substrates. The lower thermal expansion coefficient of SiO_2_ can lead to smaller curvatures during the high-temperature MOVPE process, which helps achieve better wavelength uniformity on large-sized wafers. Moreover, SiO_2_ can be easily removed by hydrofluoric acid without affecting the epitaxial layers. Finally, PECVD-produced SiO_2_ has the advantage of lower fabrication costs. The aforementioned features are beneficial for the development of SiO_2_-based micro LEDs. Indeed, the smaller refractive index of SiO_2_ (compared t Al_2_O_3_) can help improve the LEE owing to its effect on the light travel along the normal direction.

Furthermore, the variation of the nitridation time from 60 s (sample 1) to 150 s (sample 7) reveals some differences in the observed images, confirming the assumption stated above regarding the bond energy. Covalent bonds would not be broken off without enough energy in a single impact, no matter how long the impact persisted. 

The 1000 nm of sample 8 shows the pits of different states, i.e., dark dots for the deep ones and light dots for the shallow ones. It took longer for the lattice to reach the fully relaxed state, implying that the nitridation can help reduce the time and cost in mass production, which is similar to the effect on the normal PSS.

To estimate the effects of the different sputtering conditions, the complete LED epitaxial structures were made following the architecture shown in Figure 3. Figure 4a shows the light-enhanced extent by the composited PSS. The peak intensity of the sample using the 50 W/60 s condition is ~17% larger than that of the normal PSS with the sputtering AlN film using the same condition, resulting in a ~5% enhancement of the integral intensity. Using the 70 W/60 s condition seems slightly excessive, causing the enhancement of the peak intensity to decrease by ~7%. The relative performances of the 50 W/60 s, 70 W/60 s, and the normal PSS cases are matched by their corresponding HD-XRD ω mode rocking curve scanning results of (0002) and (1102) in Figure 5a and Figure 6a, respectively. The value of the (0002) FMHW is associated with the screw dislocation density and that of the (1102) FMHW is associated with the composite dislocation density. For the 60 W and 70 W cases on composited PSS, the (0002) FMHWs are 144 arcsec and 195 arcsec, respectively, the (1102) FMHWs are 191 arcsec and 229 arcsec, and the FMHWs of the normal PSS are 247 arcsec and 249 arcsec, respectively. However, the (0002) and (1102) FMHWs of the sample without nitridation are 219 arcsec and 236 arcsec, better than those of the normal PSS. This is mainly due to the absence of GaN growth on the SiO_2_ without nitridation, which enhances the ELOG process. The FMHW enlarges fast with the increase in AC power. The (0002) and (1102) FMHWs attain 368 arcsec and 312 arcsec for the 90 W/s case, while those for the 120 W/60 s case reach 614 arcsec and 687 arcsec, respectively. This observation corroborates with the previously shown SEM images. 

Figure 4b shows that the peak intensity is weakened by 32% and the integral intensity is weakened by 12% as the nitridation time increases. The (0002) FMHW increases from 144 arcsec to 219 arcsec, and the (1102) FMHW increases from 191 arcsec to 231 arcsec, as shown in Figure 5b and Figure 6b, respectively. This perhaps implies that the light weakening was mainly due to the Shockley−Read−Hall recombination and the stress relaxing rates were almost the same. It is noteworthy that the PL result of the 50 W/150 s condition is the worst in Figure 4b, whereas its corresponding XRD data are fairly good. This implies that the EQE took the disadvantages of some other factors induced by the long nitridation time. Unraveling the exact reason for this behavior necessitates further studies by designing more comparable experiments.

## 4. Conclusions

In this study, we showed results for the combination of the composited SiO_2_-PSS and the sputtering AlN films. Our findings indicate that the nitridation conditions used before sputtering affect the epitaxial results of GaN to a great extent. When the alternating current power exceeded a certain strength, the SiO_2_ was partially broken and the GaN could grow on it with the AlN buffer. Although this result is not considered good for the Ga-face epitaxy on the normal substrates, it, fortunately, provides an insight into the potential use of pure SiO_2_ substrates, which might be intrinsically more advantageous for manufacturing vertical structure chips (i.e., micro-LED and DUV LED) and low-light divergence angle chips (i.e., mini-LED and vertical-cavity surface-emitting lasers). By adjusting the AC power and duration to appropriate ranges, the composited PSS can enhance the light output by reducing the dislocation density. 

## Figures and Tables

**Figure 1 materials-16-01104-f001:**
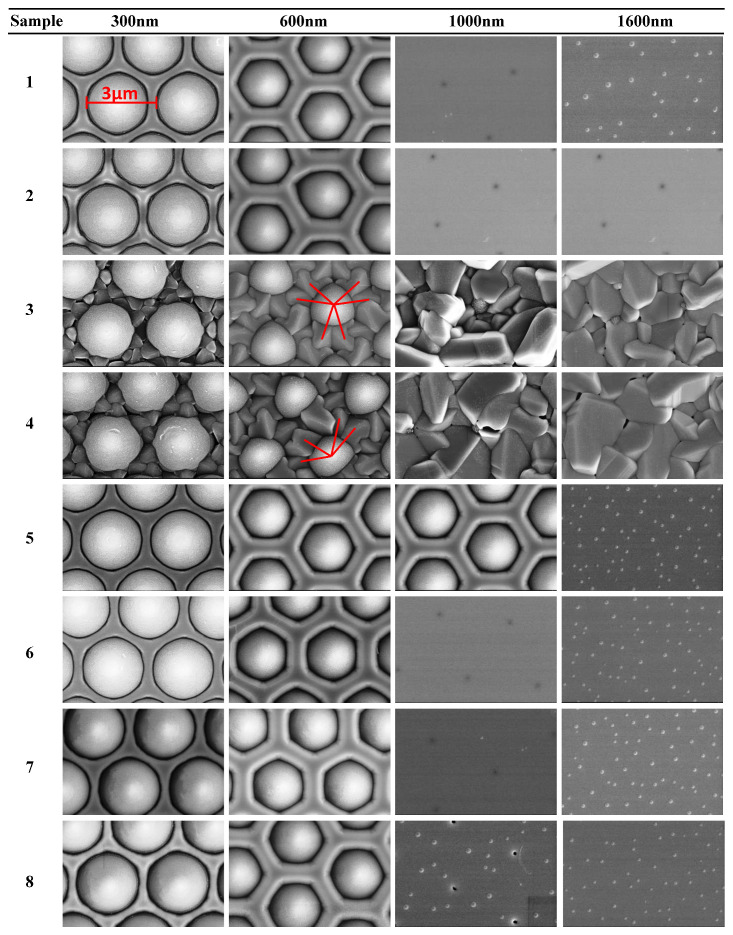
The SEM images with the magnification of 20 k. The numbers of samples match those in Table 1. All the images use the same length scale shown in the 300 nm of sample 1. The short red lines signal the growth directions of GaN on the sides of SiO_2_ patterns when the AC power is too large.

**Figure 2 materials-16-01104-f002:**
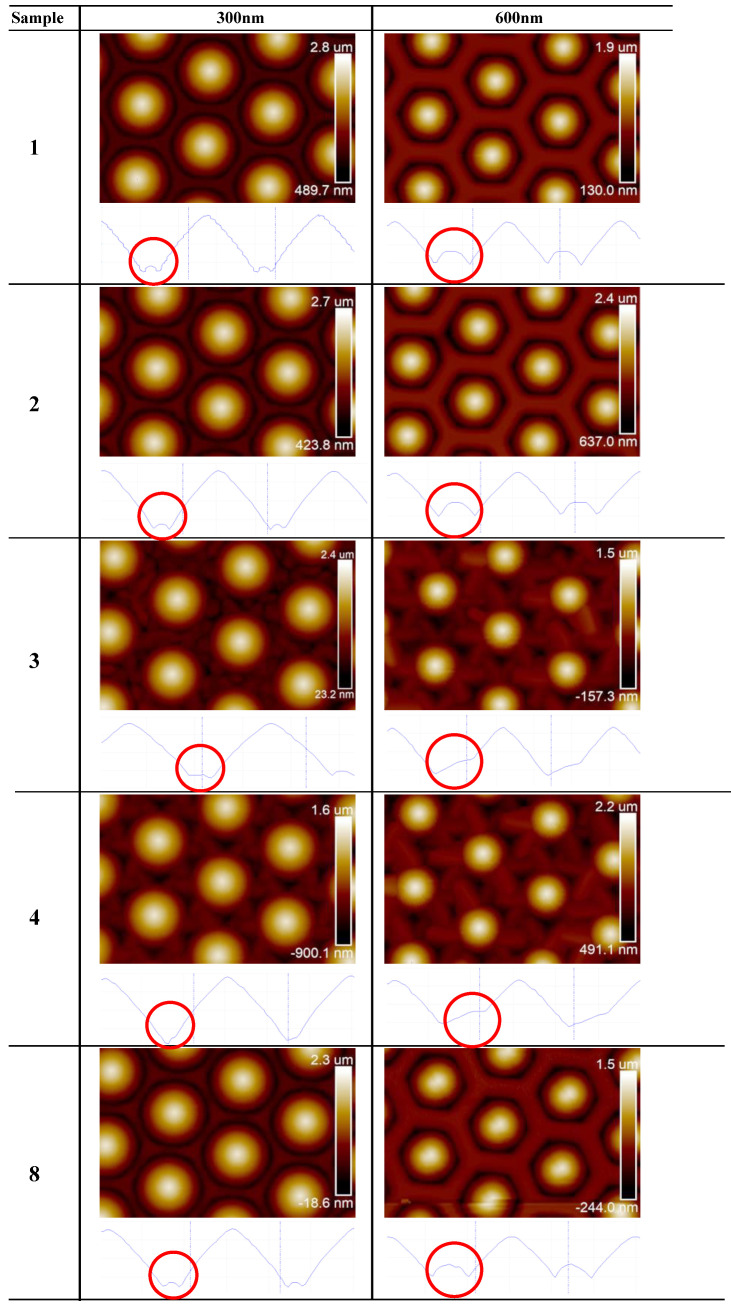
The samples contain different AC powers used in nitridation process of PVD recorded by the AFM vertical views and sections. The most evident distinctions occurred at the pedestals of the patterns pointed out by the red circles.

**Figure 3 materials-16-01104-f003:**
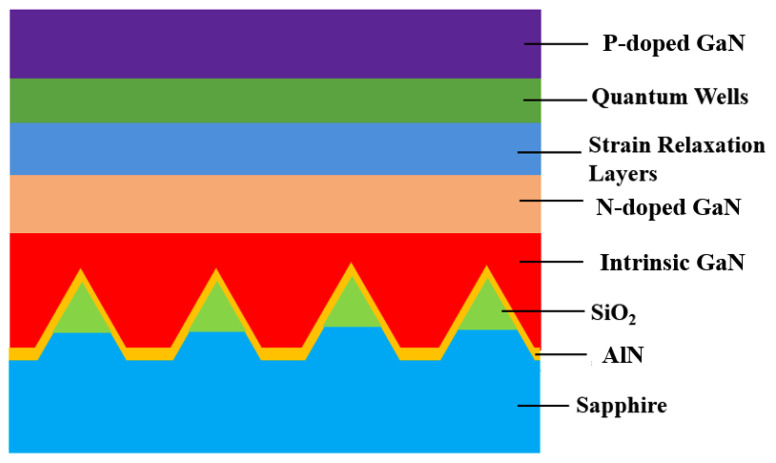
The complete epitaxial structure of LED. The illustration only reveals the locations. The size of each part does not represent the real proportion.

**Figure 4 materials-16-01104-f004:**
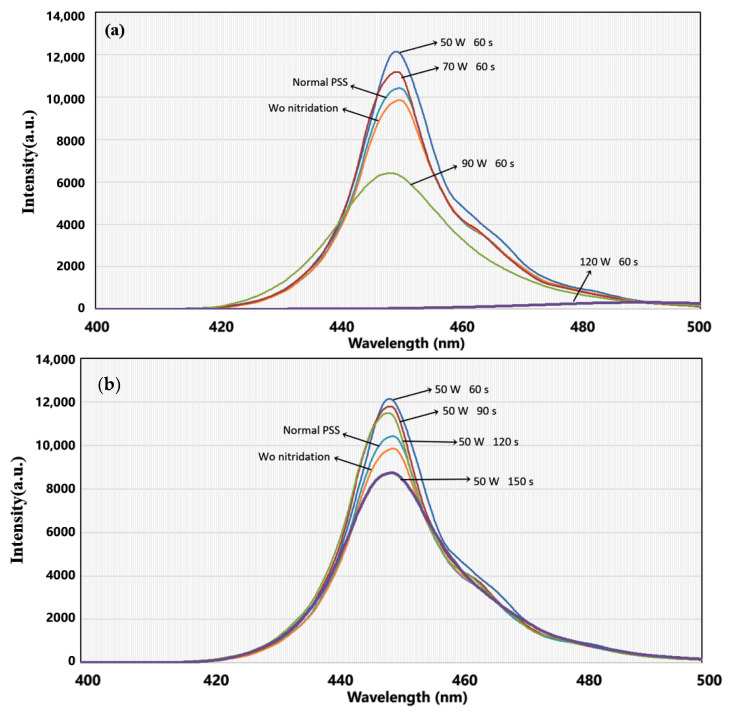
The PL spectra of the complete LED epitaxial structures classified by (**a**) the AC powers and (**b**) the nitridation times.

**Figure 5 materials-16-01104-f005:**
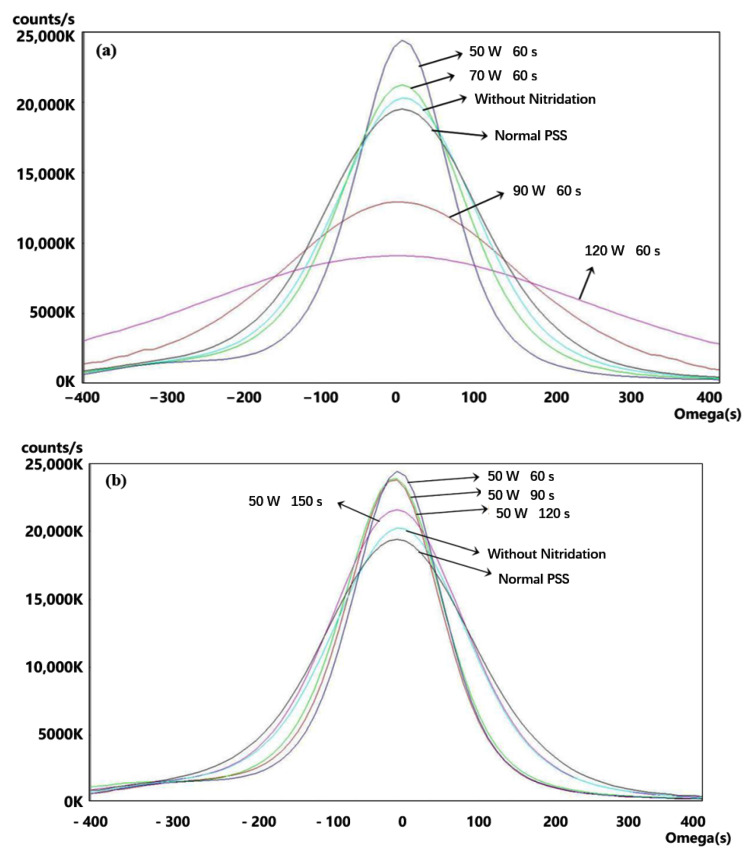
The ω mode scanning of (0002) planes on the samples of the complete LED epitaxial structure. The data were smoothed with the same order. (**a**) Results classified by the AC powers and (**b**) results classified by the nitridation times.

**Figure 6 materials-16-01104-f006:**
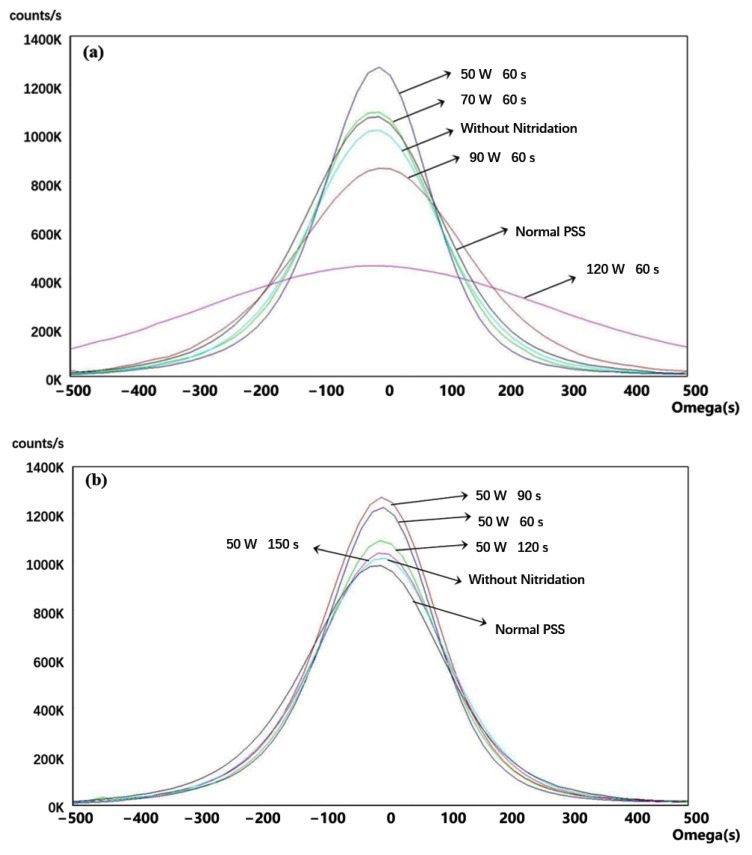
The ω mode scanning of (1102) planes on the samples of the complete LED epitaxial structure. The data were smoothed with the same order. (**a**) Results classified by the AC powers and (**b**) results classified by the nitridation times.

**Table 1 materials-16-01104-t001:** The eight deposition conditions of PVD AlN films. The thicknesses and the refractive indices belong to the films on the flat (without patterns) c-plane sapphire substrates tested by ellipsometry.

Sample	AC Power (W)	Nitridation Time (s)	Thickness (nm)	Refractive Index
1	50	60	11.51	1.9706
2	70	60	11.51	1.9686
3	90	60	11.43	1.9707
4	120	60	11.55	1.9703
5	50	90	11.31	1.9670
6	50	120	11.34	1.9646
7	50	150	11.44	1.9649
8	/	/	11.7	1.9811

## Data Availability

The data is not available for it’s related to some confidential business.

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
