# Peer review of "The Effect of Nitridation on Sputtering AlN on Composited Patterned Sapphire Substrate"

_materials, 2023, doi:10.3390/ma16031104_

Round 1

Reviewer 1 Report

  1. One of the authors' observations, apart from the above-mentioned ones, was to reduce the number of screw dislocations, affecting, as we know, the lifetime of the laser diode. It is a pity that the authors did not conduct appropriate resource studies. Authors stated, that the new patterned supphire substrate design can enhance the epitaxial lateral overgrowth process, eliminating the dislocations by changing the trends of dislocation lines. I could not understand how. In blue leds it was done by appropriate codoping with e.g. In ions. Please explain it more clearly. 

  1. I can not understand the following sentence: “By replacing the material of the cones with silicon oxide (n = 1.45), the LEE can be further enhanced as it becomes harder for the light to penetrate through the substrate (Is the material harder for the light? What does it mean? Does refractive index increase?) 

  1. To avoid interface reflections, resulting from refractive index mismatch there are applied substrates with a given curvature. Is it something similar in this case? 

  1. Why the work proposes a sapphire substrate and not a single GaN crystal? 

  1. Nevertheless performed experiments are very interesting 

  1. Some abbreviations, so eagerly used by authors, were used before explaining their meaning, e.g. PVD. 

  1. Figures 4-6 are hardly illegible. Too small digits on the ordinate and abscissa axes.

Author Response

1、Authors stated, that the new patterned supphire substrate design can enhance the epitaxial lateral overgrowth process, eliminating the dislocations by changing the trends of dislocation lines. I could not understand how. In blue leds it was done by appropriate codoping with e.g. In ions. Please explain it more clearly. 

Response:

The epitaxial lateral overgrowth process is very necessary to the elimination of screw dislocations which starts from the c-plane of sapphire. When there is no pattern on the sapphire surface, the screw dislocations extend vertically through the GaN layer, leading to the electron leakage passages and nonradiative recombination centers. The incline surfaces of PSS hardly produce screw dislocations and force the dislocation lines to turn into tilt directions, confluence and vanish finally.

The above explanation is added to the manuscript.

2、I can not understand the following sentence: “By replacing the material of the cones with silicon oxide (n = 1.45), the LEE can be further enhanced as it becomes harder for the light to penetrate through the substrate (Is the material harder for the light? What does it mean? Does refractive index increase?) To avoid interface reflections, resulting from refractive index mismatch there are applied substrates with a given curvature. Is it something similar in this case? 

Response:The LEE promotion is achieved with the total reflection enhancement. The n value of silicon oxide is smaller than that of sapphire, leading to more total reflections at the interface between SiO2 and sapphire. It’s a very significant effect for the tilt light to be reflected to the PGaN on top of epitaxial layers instead of penetrating through the sapphire substrate.

The above explanation is added to the manuscript.

3、Why the work proposes a sapphire substrate and not a single GaN crystal? 

Response: This work proposes sapphire substrate instead of single GaN crystal mainly because of the industrial cost. Considering of the large-scale application in GaN based LEDs, the sapphire substrate takes more advantages in cost than GaN substrate. Furthermore, the GaN homoepitaxy doesn’t need PVD AlN film as buffer.

4、Some abbreviations, so eagerly used by authors, were used before explaining their meaning, e.g. PVD. 

Response:Sorry for the carelessness. The mistakes are corrected in our new uploaded manuscript.

5、Figures 4-6 are hardly illegible. Too small digits on the ordinate and abscissa axes.

Response:The digits on axes are tuned to be larger for reading in our new uploaded manuscript.

Reviewer 2 Report

This manuscript reports on the effect of nitridation on sputtering AlN on composited patterned sapphire substrate. The research results indicate that the nitridation applied before sputtering affects the epitaxial growth of GaN very significantly when related to sputtering a buffer AlN layer. The growth schemes and their implications on the quality of the material are well transmitted by the discussion in the present manuscript. The research questions as defined are very much worth investigation while in the present context the material and demonstrator system, namely, the combination of the composited SiO2-PSS and the sputtering of AlN film are well identified, nicely motivated, specifically explained and comprehensively discussed. The characterization efforts (including HD-XRD, PL) as employed are not only adequate for the present purpose, but also the interpretation of characterization results seems correctly done.

All in all, the results are interesting and presented in a way which is easy and valuable for a wider audience of readers. All these results seem like worthy points of departure for the discussion and are plausible and well substantiated in the view of the conclusions drawn.

The reported results bring new knowledge and certainly represent an original contribution with probable technological impact in catalysis.

The authors chose an adequate structure of the manuscript for such a study. Also, they provided a balanced realistic and nicely illustrated presentation of their results and corresponding analysis that is of much scientific and practical interest and adds new knowledge to the field.

The present manuscript is a significant contribution, this work once published would be instructive and suggestive in terms of further studies and with good chances be cited.

There are some minor issues with this already excellent manuscript that will need to be addressed before the manuscript becoming suitable for publication, i.e., it can be considered for publication after a minor revision:

1: Authors are a little bit too telegraphic in what concerns the characterization methods (which is much adequate) especially in the abstract (where short mentioning of characterization techniques is missing). It will publicize their work better if they are more detailed/descriptive concerning this issue.

2: Title is not optimal. The word “condition” is obsolete. Also, prepositions are wrongly used. I would suggest something like: “The effect of nitridation on sputtering AlN on composited ….”

3: The MOVPE process temperatures should be mentioned explicitly in the text.

4: In the introduction, the authors miss that property and synthesis of composites of nitride materials is frequently guided by specially developed theoretical and ab initio methodology especially regarding metal-organic chemistry of nitrides synthesis [e.g., Dalton Transactions 44 (2015) 3356-3366, and CrystEngComm 23 (2021) 385-390] with direct practical implications for the credibility of the claims of the present manuscript. This aspect should be acknowledged in the present manuscript.

5: Spell-check and stylistic revision of the paper are necessary. Some long sentences, wrong word order, inappropriate expressions/terminology, as well as misspellings, etc., are noticeable throughout the text.

Author Response

1: Authors are a little bit too telegraphic in what concerns the characterization methods (which is much adequate) especially in the abstract (where short mentioning of characterization techniques is missing). It will publicize their work better if they are more detailed/descriptive concerning this issue.

Response: Thank you for the advice. We enriched the description of AFM, SEM, PL and XRD test in abstract in our new uploaded manuscript. Please see the attachment.

2: Title is not optimal. The word “condition” is obsolete. Also, prepositions are wrongly used. I would suggest something like: “The effect of nitridation on sputtering AlN on composited ….”

Response: Thank you for the advice. We changed our manuscript title as you suggested. Please see the attachment.

3: The MOVPE process temperatures should be mentioned explicitly in the text.

Response:The MOVPE process temperatures of PN junction growth are added in our new uploaded manuscript. Please see the attachment.

4: In the introduction, the authors miss that property and synthesis of composites of nitride materials is frequently guided by specially developed theoretical and ab initio methodology especially regarding metal-organic chemistry of nitrides synthesis [e.g., Dalton Transactions 44 (2015) 3356-3366, and CrystEngComm 23 (2021) 385-390] with direct practical implications for the credibility of the claims of the present manuscript. This aspect should be acknowledged in the present manuscript.

Response:Thank you for the advice. We add the theoretical work references in our new uploaded manuscript. Please see the attachment.

5: Spell-check and stylistic revision of the paper are necessary. Some long sentences, wrong word order, inappropriate expressions/terminology, as well as misspellings, etc., are noticeable throughout the text.

Response: Thank you for the advice. We let the manuscript be polished by professional agency and correct the wrong expression. Please see the attachment.
